# The Pregnancy-Unique Quantification of Emesis and Nausea (PUQE-24): Configural, Measurement, and Structural Invariance between Nulliparas and Multiparas and across Two Measurement Time Points

**DOI:** 10.3390/healthcare9111553

**Published:** 2021-11-15

**Authors:** Ayako Hada, Mariko Minatani, Mikiyo Wakamatsu, Gideon Koren, Toshinori Kitamura

**Affiliations:** 1Kitamura Institute of Mental Health Tokyo, Flat A, Tomigaya Riverland House, 2-26-3 Tomigaya, Shibuya, Tokyo 151-0063, Japan; hada@institute-of-mental-health.jp; 2Kitamura KOKORO Clinic Mental Health, Tokyo 151-0063, Japan; 3Life Value Creation Unit, NTT DATA Institute of Management Consulting, Inc., Tokyo 102-0093, Japan; mrk.minatani0405@gmail.com; 4Department of Reproductive Health Care Nursing, School of Health Sciences, Faculty of Medicine, Kagoshima University, Kagoshima 890-8544, Japan; mikiwaka@health.nop.kagoshima-u.ac.jp; 5Adelson Faculty of Medicine, Ariel University, Ariel 40700, Israel; Gidiup_2000@yahoo.com; 6Motherisk Israel, Aiel University, Tel Aviv 47000, Israel; 7T. and F. Kitamura Foundation for Studies and Skill Advancement in Mental Health, Tokyo 151-0063, Japan; 8Department of Psychiatry, Graduate School of Medicine, Nagoya University, Nagoya 464-8601, Japan

**Keywords:** PUQE-24, factor structure, measurement and structural invariance, parity, two observation occasions

## Abstract

Background: The severity of nausea and vomiting of pregnancy (NVP) correlates with pregnancy complications. This study aimed to confirm the measurement and structural invariance of the 24 h Pregnancy-Unique Quantification of Emesis and Nausea (PUQE-24) regarding parity and observation time among pregnant women during the first trimester. Methods: Questionnaires including the PUQE-24 and the Health-Related Quality of Life for Nausea and Vomiting during Pregnancy (NVP-QOL) questionnaire were distributed to pregnant women from 10 to 13 weeks of gestation who were attending antenatal clinics. There were 382 respondents, and of these, 129 responded to the PUQE-24 again one week later. Results: Confirmatory factor analysis of this single factor model showed a good fit with the data: CFI = 1.000. The PUQE-24 factor and NVP-QOL factor were strongly correlated (*r* = 82). Configural, measurement, and structural invariance of the factor structure of the PUQE items were confirmed between primiparas and multiparas as well as at the test and retest observation occasions. Conclusion: The findings suggested that using the PUQE-24 among pregnant women in the first trimester was robust in its factor structure. The PUQE-24 may be a promising tool as an easy and robust measure of the severity of nausea and vomiting among pregnant women.

## 1. Background

Nausea and vomiting of pregnancy (NVP) is a common health issue among pregnant women. NVP involves any degree or duration of nausea with or without vomiting or retching and is not associated with other known causal factors. Almost 70% of women worldwide experience NVP, with higher rates reported in East Asian countries [1]. Nutritional disturbances, weight loss, dehydration, and ketonuria may lead to hospitalisation [2]. Hyperemesis gravidarum (HG) is considered the most severe form of NVP. Einarson et al. [1] reported that the prevalence of HG was 1.1%. In addition to the emergence of somatic symptoms, a woman’s quality of life (QOL) and ability to function are also impaired among those who suffered from severe NVP [3,4]. It is also associated with high blood pressure [5], having a low birth weight infant, an infant small for gestational age, or a preterm delivery [6,7] and depression, post-traumatic stress disorder, as well as anxiety disorders [8,9]. For clinical and research use, a reliable and valid, as well as simple, measure for quantification of NVP severity is required.

The Pregnancy-Unique Quantification of Emesis and Nausea (PUQE; Koren et al. 2002) [10] is a severity measure used in studies to determine the burden or treatment outcome of NVP. The PUQE is a scoring system for nausea and vomiting during pregnancy, which consists of three items. The PUQE was developed for pregnant women on the basis of the Rhodes Index of Nausea and Vomiting (INV; Rhodes et al. 1984) [11] and focuses on three symptoms: nausea, vomiting, and retching. The original PUQE entailed rating the daily number of vomiting episodes, the length of nausea in hours per day, and the number of retching episodes per 12 h. Its validation was confirmed by Koren et al. [12]. To capture more comprehensive NVP severity, the PUQE was modified by Lacasse et al. [13]. The Modified-PUQE (PUQE-24) is a scoring system per 24 h with the same scoring calculation and interpretation as for the original PUQE. The PUQE is widely used as a scoring system to assess NVP severity in many countries [14,15,16,17]. Its use as the validated tool should be applied more frequently in better defining the severe end of HG [18].

The present study shows the psychometric properties of the PUQE-24 among pregnant women, including confirmatory factor analysis (CFA) and configural, measurement, and structural invariance of the factor structure. We focused on the invariance of the factor structure between nulliparas and multiparas and between the test and retest occasions. When a psychological instrument is used in different populations or used at more than one measurement occasion, both selection of the best fit model of factor structure and confirmation of the postulation that the psychological instrument in question measures the same phenomena is needed. If this confirmation is not achieved, the instrument does not reflect the same phenomenon, and the results may be biased. Invariance tests take several steps [19]. First, each group (e.g., nulliparas vs. multiparas) has the same pattern of the indicators and factors (configural invariance). Second, factor loadings for like indicators are invariant across groups (metric invariance; also known as weak factorial invariance). Third, intercepts of like items are invariant across groups (scalar invariance; also known as strong factorial invariance). Fourth, residuals (errors) of like items are invariant across groups (residual invariance; also known as strict factorial invariance). Fifth, variances of like factors are invariant across groups (factor variance invariance). Sixth, means of factors are invariant across groups (factor mean invariance). The second to fourth steps are called measurement invariance. The fifth and sixth steps are called structural invariance [19]. If one step is rejected, the next steps cannot be performed. We conducted tests for our hypotheses using this algorithm.

On the procedure of the data analysis and explanation, we followed the Consensus-based Standards for the selection of health Measurement Instruments (COSMIN) Study Design checklist [20]. It is recommended for designing studies and evaluating measurement properties, including content validity, structural validity, internal consistency, cross-cultural validity/measurement invariance, reliability, measurement error, criterion validity, hypotheses testing for construct validity, and responsiveness [21,22]. The PUQE-24 has been evaluated its content validity [23], hypothesis testing for construct validity [24,25] and criterion validity [14,23]. To the best of our knowledge, its validity, including the measurement invariance, has not been examined.

## 2. Methods

### 2.1. Study Procedures and Participants

This study was a longitudinal follow-up conducted at a 1-week interval because of the goal to examine measurement invariance between the test and retest occasions. We solicited approximately 1500 pregnant women at 10 to 13 weeks of gestation at the antenatal clinic of two general hospitals and four private clinics located in Tokyo, Chiba, Ibaraki, and Kagoshima Prefectures in Japan. There were 382 total respondents (approximately 25% of those solicited). They were provided with a set of test and retest questionnaires and were asked to return the retest questionnaire 1 week later. Of the respondents, 129 sent back the retest questionnaire. Voluntary participation and anonymity were assured. Responses for the two time occasions were matched by a predetermined number on the questionnaire. Pregnant women were excluded if they: (a) were not fluent in Japanese, (b) were aged under 20, (c) had eating disorders, (d) had symptoms of vaginal bleeding or abdominal pain, (e) had a subchorionic haematoma, or (f) experienced recurrent miscarriages. The mean (SD) age of the participants was 31.9 (4.9) years old, and the mean (SD) age of their partners was 33.5 (5.5) years old. Most of them were engaged in a relationship (94.5%). Of the respondents, 44.0% were nulliparas, and 55.0% were multiparas. This was a convenience sample. However, the sample consisted of those women receiving different types of obstetrical services in Japan. Data collection was conducted from January 2017 to May 2019.

### 2.2. Measurements

The 24 h Pregnancy-Unique Quantification of Emesis and Nausea (PUQE-24; Ebrahimi et al. 2009) [23] is a self-measure rating (a) nausea (the length of nausea in hours for the last 24 h), (b) vomiting (number of vomiting episodes in the last 24 h), and (c) retching (the number of retching episodes in the last 24 h) each with a 5-point scale. Higher scores indicate more severe NVP. The PUQE-24 was translated into Japanese by M.M. and T.K. with permission from the original authors. The bilingual author of the original version back-translated it and compared it with the original English.

The Japanese version [26] of the Health-Related Quality of Life for Nausea and Vomiting during Pregnancy questionnaire (NVP-QOL; Magee et al. 2002) [27] was used simultaneously as a measure of concurrent validity. This has 30 items with a 7-point scale measuring NVP and related QOL in the previous week. Higher scores indicate more severe NVP and worse QOL. The NVP-QOL has a single factor structure [26].

### 2.3. Data Analysis

The PUQE-24 should have a single-factor structure because it consists of only three items. Hence, we only performed exploratory factor analyses (EFAs) for a single-factor structure. After calculating the mean, SD, skewness, and kurtosis of each PUQE-24 item, we examined a factorability check of the PUQE-24 with the Keiser-Meyer-Olkin (KMO) index and Bartlett’s sphericity test [28]. A single-factor EFA derived the factor loading of the PUQE-24 items. Then, a CFA of the single-factor analysis was examined and checked for its goodness-of-fit. The fit of the models was checked in terms of chi-squared and comparative fit index (CFI). A good fit would be indicated by *χ*^2^/*df* < 2, and CFI > 0.97, and an acceptable fit by *χ*^2^/*df* < 3, and CFI > 0.95 [29,30].

The model’s configural, measurement, and structural invariance were examined across parity and observation occasions. Starting from the configural invariance, we went through metric, scalar, residual, and factor variance invariances to factor mean invariances [19,31]. The progress from one step to the next was judged as ‘accepted’ if (a) the *χ*^2^ decrease was not significant for the *df* difference, (b) the decrease of CFI was less than 0.01, or (c) the increase of root mean square of error approximation (RMSEA) was less than 0.01 [32,33]. We applied this procedure because a *χ*^2^ decrease is strongly sensitive to the sample size (*N*) and, particularly in the case of a large sample, produces an unreasonable rejection of invariance.

## 3. Results

Mean, SD, skewness, and kurtosis of the three PUQE-24 items are in Table 1. One item showed a slightly high skewness (2.93) and kurtosis (9.68). The KMO was 0.625, and Bartlett’s sphericity was 190.796 (3) (*p* < 0.001). Therefore, the data appeared factorable. Factor loading of each item in the single-factor model is in Table 1. This model explained 61% of the whole variance. Confirmatory factor analysis of this single factor model showed a good fit with the data: CFI = 1.000. The PUQE-24 factor and the NVP-QOL factor were strongly correlated (*r* = 0.82).

Configural and measurement invariances are accepted between primiparas and multiparas as well as between the test and retest occasions (Table 2). Factor mean also did not differ between primiparas and multiparas as well as between the test and retest occasions (Table 3).

## 4. Discussion

The present study showed that the single-factor structure of the PUQE-24 was robust among pregnant Japanese women. Its structure was invariant regardless of parity as well as observation times. Concurrent validity with the NVP-QOL scores was also excellent. According to Ebrahimi et al. [23], the PUQE-24 has exactly reflected pregnant women’s severity of symptoms of NVP during one day. Taking into account the PUQE-24′s simplicity, we think that the use of the PUQE-24 in clinical and research settings in antenatal maternal care is extremely promising. This is particularly the case when clinicians and researchers wish to distinguish between cases of severe NVP, most likely due to HG, and mild and moderate cases. The PUQE-24 may be used as an outcome measure of intervention by midwives and other perinatal health professionals. The correlation we found between emesis severity and quality of life may lead to further study of the biological relationship between the NVP and the outcome of pregnancy.

There are several limitations to this study. Our study sample size was medium and based on a convenience sample. The participation rate was approximately 25%. Hence the results may have been biased. Nevertheless, the participation rate of epidemiological studies among non-clinical populations in Japan is usually as low as one in four. It may be that those women with few emesis symptoms were not interested in participating in this study and therefore declined. Although we had better compare those women who participated and those who did not in terms of major variables used in this study, it was ethically not permitted. One of the inclusion criteria was pregnant women at 10 to 13 weeks of gestation. Though we intended to have a homogeneous population of pregnant women for this study sample, different results may have been produced if women at different weeks of gestation had been studied. Yet, the range of 10–13 weeks gestation is very slim and not likely to increase variability. Hence, careful generalisation is needed. The findings were based on self-reporting. Further examination of the degree of concordance between their reports and clinical observers’ or family members’ reports is needed.

Taking these drawbacks into consideration, the PUQE-24 appears to be a promising tool as an easy and robust measure of the severity of NVP among pregnant women.

## 5. Conclusions

The findings suggested that using the PUQE-24 among pregnant women in the first trimester was robust in its factor structure. The PUQE-24 may be a promising tool as an easy and robust measure of the severity of nausea and vomiting among pregnant women.

## Figures and Tables

**Table 1 healthcare-09-01553-t001:** Mean, SD, skewness, and kurtosis of PUQE-24 items (*n* = 378).

	ITEM	*n*	Mean	SD	Skewness	Kurtosis	Factor Loading of 1-Factor Model
1	In the last 24 h, how long have you felt nauseated or sick to your stomach?	377	3.1	1.5	0.00	−1.31	0.77
2	In the last 24 h, have you vomited or thrown up?	378	1.3	0.6	2.93	9.68	0.46
3	In the last 24 h, how many times have you had retching or dry heaves without bringing anything up?	378	2.1	1.3	1.00	−0.02	0.70

**Table 2 healthcare-09-01553-t002:** Measurement and structural invariance of the PUQE-24.

	χ^2^	*df*	χ^2^/*df*	△χ^2^ (*df*)	CFI	△CFI	RMSEA	△RMSEA	Judgement
Nulliparas (*n* = 168) vs. Multiparas (*n* = 210)									
Configural	0.000	0	0	Ref	1.000	Ref	0.000	Ref	ACCEPT
Metric	0.949	2	0.474	0.949 (2) **	1.000	0.000	0.000	0.000	ACCEPT
Scalar	3.788	5	0.758	2.840 (3) *	1.000	0.000	0.000	0.000	ACCEPT
Residual	4.325	8	0.541	0.540 (3) NS	1.000	0.000	0.000	0.000	ACCEPT
Factor variance	4.356	9	0.484	0.027 (1) NS	1.000	0.000	0.000	0.000	ACCEPT
Time 1 (*n* = 382) vs. Time 2 (*n* = 129)									
Configural	0.000	0	0	Ref	1.000	Ref	0.000	Ref	ACCEPT
Metric	1.089	2	0.545	1.089 (2) NS	1.000	0.000	0.000	0.000	ACCEPT
Scalar	2.940	5	0.588	1.851 (3) NS	1.000	0.000	0.000	0.000	ACCEPT
Residual	9.211	8	1.151	6.271 (3) NS	0.995	0.005	0.017	0.017	ACCEPT
Factor variance	10.367	9	1.152	1.089 (2) NS	0.994	0.001	0.017	0.000	ACCEPT

* *p* < 0.05; ** *p* < 0.01; NS, not significant.

**Table 3 healthcare-09-01553-t003:** Factor mean invariance of the PUQE-24.

Comparison	Differences in Factor Mean (SE)
Multiparas as compared with nulliparas	0.148 (0.103) NS
Time 2 as compared with Time 1	−0.136 (0.105) NS

NS, not significant; SE, standard error.

## Data Availability

Data used in this study will be obtained upon reasonable request to the corresponding author.

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
