# Peer review of "The Pregnancy-Unique Quantification of Emesis and Nausea (PUQE-24): Configural, Measurement, and Structural Invariance between Nulliparas and Multiparas and across Two Measurement Time Points"

_healthcare, 2021, doi:10.3390/healthcare9111553_

Round 1

Reviewer 1 Report

Reading the article in its introductory part leads to believe
that the use of the proposed questionnaire will give important
information on the therapeutic conduct to be followed. However,
this is not taken into account in the discussion. Despite the importance
of symptom quantification that is possible with the questionnaire,
there is no indication for treatment based on the severity of the disease.
Excellent correlation between symptom severity and quality of life leads to further
sstudy the biological relationship between the NVP and the outcome of pregnancy..

Author Response

REVIEWER #1

Reading the article in its introductory part leads to believe that the use of the proposed questionnaire will give important information on the therapeutic conduct to be followed. However, this is not taken into account in the discussion. Despite the importance of symptom quantification that is possible with the questionnaire, there is no indication for treatment based on the severity of the disease. Excellent correlation between symptom severity and quality of life leads to further study the biological relationship between the NVP and the outcome of pregnancy.

The reviewer’s comment is succinct. We deleted and revised Introduction so that it would not give an impression that would be linked to suggestion about treatment of emesis.

Nausea and vomiting of pregnancy (NVP) is a common health issue among pregnant women. NVP involves any degree or duration of nausea with or without vomiting or retching and which is not associated with other known causal factors. Almost 70% of women worldwide experience NVP, with higher rates reported in East Asian countries [1]. Nutritional disturbances, weight loss, dehydration, and ketonuria may lead to hospitalisation [2]. Hyperemesis gravidarum (HG) is considered the most severe form of NVP. Einarson et al. [1] reported that the prevalence of HG was 1.1%. In addition to the emergence of somatic symptoms, a woman’s quality of life (QOL) and ability to function are also impaired among those who suffered from severe NVP [3, 4]. It is also associated with high blood pressure [5], having a low birth weight infant, an infant small for gestational age, or a preterm delivery [6, 7] and depression, post-traumatic stress disorder, as well as anxiety disorders [8, 9]. For clinical and research use, a reliable and valid, as well as simple, measure for quantification of NVP severity is required.

We also added the following sentence at the end of the first paragraph in Discussion.

Correlation we found between emesis severity and quality of life may lead to further study the biological relationship between the NVP and the outcome of pregnancy.

Reviewer 2 Report

This is a multi-centre questionnaire-based/ study done in 129 participants who rated their nausea, vomiting and retching symptoms over a 24 hour period between gestational weeks 10-13. The PUQE-24 questionnaire which is a short, 3 question scoring system was used to determine the severity of symptoms along with HR-QOL score for women with hyper-emesis gravidarum. Both these tools are used for screening in women with NVP and are well described in literature as directly applicable to the target population and demonstrating overall consistency of results.

General comments:

  1. The authors report screening 1500 women over an almost 2 year period, but only approx. 25% patients participated in the short survey for the test and only 9% reverted with the retest- what do the authors attribute this minimal number of participations to?
  2. Is it likely that due to the small cohort size for this study , the results would not be generalizable?
  3. Is it possible that the number of women with symptoms in this cohort were low and therefore did not find it useful to participate?
  4. The interpretation of the results should be explained in simpler terms. The statistical interpretations are complex and need more description if the object of the manuscript is to reach a wider audience.
  5. What do the authors think this particular study adds to the already existing literature on scoring systems, since the PUQE and QOL are already validated for use in this targeted population?

Author Response

REVIEWER #2

The authors report screening 1500 women over an almost 2 year period, but only approx. 25% patients participated in the short survey for the test and only 9% reverted with the retest- what do the authors attribute this minimal number of participations to? Is it likely that due to the small cohort size for this study, the results would not be generalizable? Is it possible that the number of women with symptoms in this cohort were low and therefore did not find it useful to participate?

We revised the first few sentences in the paragraph describing the limitation of the study as follows.

There are several limitations to this study. Our study sample size was medium and based on a convenience sample. The participation rate was approximately 25%. Hence the results may have ben biased. Nevertheless, the participation rate of epidemiological studies among non-clinical populations in Japan is usually as low as one in four. It may be that those women with few emesis symptoms were not interested in participating in this study and therefore declined.

The interpretation of the results should be explained in simpler terms. The statistical interpretations are complex and need more description if the object of the manuscript is to reach a wider audience. What do the authors think this particular study adds to the already existing literature on scoring systems, since the PUQE and QOL are already validated for use in this targeted population?

We described the outline and importance of testing strict psychometric properties in Introduction (see below).

When a psychological instrument is used in different populations or used at more than one measurement occasion, both selection of the best fit model of factor structure and confirmation of the postulation that the psychological instrument in question measures the same phenomena is needed. If this confirmation is not achieved, the instrument does not reflect the same phenomenon and the results may be biased.

the PUQE and QOL are already validated for use in this targeted population. Nevertheless, what we wanted to emphasise is its lack of validating measurement invariance (see below).

To the best our knowledge, its validity including the measurement invariance has not been examined.

REVIEWER #2

The authors report screening 1500 women over an almost 2 year period, but only approx. 25% patients participated in the short survey for the test and only 9% reverted with the retest- what do the authors attribute this minimal number of participations to? Is it likely that due to the small cohort size for this study, the results would not be generalizable? Is it possible that the number of women with symptoms in this cohort were low and therefore did not find it useful to participate?

We revised the first few sentences in the paragraph describing the limitation of the study as follows.

There are several limitations to this study. Our study sample size was medium and based on a convenience sample. The participation rate was approximately 25%. Hence the results may have ben biased. Nevertheless, the participation rate of epidemiological studies among non-clinical populations in Japan is usually as low as one in four. It may be that those women with few emesis symptoms were not interested in participating in this study and therefore declined.

The interpretation of the results should be explained in simpler terms. The statistical interpretations are complex and need more description if the object of the manuscript is to reach a wider audience. What do the authors think this particular study adds to the already existing literature on scoring systems, since the PUQE and QOL are already validated for use in this targeted population?

We described the outline and importance of testing strict psychometric properties in Introduction (see below).

When a psychological instrument is used in different populations or used at more than one measurement occasion, both selection of the best fit model of factor structure and confirmation of the postulation that the psychological instrument in question measures the same phenomena is needed. If this confirmation is not achieved, the instrument does not reflect the same phenomenon and the results may be biased.

the PUQE and QOL are already validated for use in this targeted population. Nevertheless, what we wanted to emphasise is its lack of validating measurement invariance (see below).

To the best our knowledge, its validity including the measurement invariance has not been examined.

Round 2

Reviewer 2 Report

acceptable changes have been made